# Network analysis of academic achievement and psychological characteristics of secondary school adolescents

Esin Yılmaz Koğar[1], Ayfer Sayın[2], Hakan Koğar[3] , Hüseyin Kafes[3] and Güçlü Şekercioğlu[3]

[1]Faculty of Education, Niğde Ömer Halisdemir University, Niğde, Türkiye; [2]Faculty of Education, Gazi University, Ankara, Türkiye and [3]Faculty of Education, Akdeniz University, Antalya, Türkiye

## Research Article

network analysis; self-efficacy; academic resilience; cognitive test anxiety; academic achievement

**Corresponding author:**
Güçlü Şekercioğlu;
Email: guclus@akdeniz.edu.tr

## Abstract

In this study, network analysis was conducted using an exploratory approach on the variables of self-efficacy, academic resilience (AR), cognitive test anxiety and academic achievement (ACH), which are frequently examined in educational research. Data were collected from a total of 828 Turkish secondary school adolescents (51.9% female), using three different self-reported scales for self-efficacy, AR and cognitive test anxiety, as well as an ACH scale. The data were analyzed using regularized partial correlation network analysis (EBICglasso). The results show that academic self-efficacy (ASE) stands out among the variables of the study and that there is a positive relationship between ASE and all other variables except cognitive test anxiety. Besides, increasing students' ASE and AR levels plays a notable role in increasing their ACH levels. By providing new evidence on the relationships among these variables, this study offers insights that may inspire educational policy interventions.

## Impact statement

The variables of self-efficacy, academic resilience (AR), cognitive test anxiety and academic achievement (ACH) – all of which are commonly studied in educational research – were the subject of an exploratory network analysis in this study. Dataset was gathered from 828 adolescents enrolled in secondary schools in Turkey. The findings indicate that academic self-efficacy (ASE) is the most connected variable of the study's variables and that it positively correlates with every other variable, with the exception of cognitive test anxiety. Additionally, increasing students' levels of AR and ASE has a significant impact on increasing their ACH levels.

## Introduction

In studies related to student achievement and psychology, variables such as self-efficacy, academic resilience (AR) and test anxiety are frequently utilized by educators and educational researchers. The relationships involving self-efficacy and achievement (Bong, 2013; Chemers et al., 2001; Pajares and Schunk, 2001; Talsma et al., 2021; Zajacova et al., 2005), AR and achievement (Chisholm-Burns et al., 2021; Kotzé and Kleynhans, 2013; Mwangi et al., 2015), test anxiety and achievement (Cassady and Johnson, 2002; Smith and Smith, 2002; Steinmayr et al., 2016), test anxiety and AR (Lei et al., 2021; Lim and Chue, 2023), test anxiety and self-efficacy (Barrows et al., 2013; Nie et al., 2011) and self-efficacy and AR (Cassidy, 2015; Martin and Marsh, 2009; Rachmawati et al., 2021; Wu et al., 2024) have been well-documented. Such research has informed educational policy. However, there is a scarcity of research exploring the relationship among these variables. In this study, we therefore aimed to examine the correlations among student self-efficacy, AR, cognitive test anxiety and academic achievement (ACH) using network analysis.

### Student self-efficacy, AR, text anxiety and academic performance

Self-efficacy is one of the core concepts of Albert Bandura's social cognitive theory, and the publication of Bandura's (1977) "Self-Efficacy: Toward a Unifying Theory of Behavioral Change" is considered as the starting point (Pajares, 1996). The construct of self-efficacy reflects an optimistic self-belief, that is, the belief in one's ability to accomplish a new or difficult task and in their capacity to have control over their own functioning (Schwarzer and Luszczynska, 2023). Zimmerman (1995) adds that the concept of self-efficacy refers to a person's awareness of his or her ability to organize and execute tasks necessary to make progress toward academic skills and

goals. In its short history of about 30 years, studies on this concept have been carried out in many different fields, such as education and psychology. This is because self-efficacy reflects how people feel, think and behave and affects both the reactions and thinking patterns of individuals.

Zimmerman (2000) stated that self-efficacy beliefs are not a single tendency; on the contrary, they exhibit multidimensional characteristics and should be evaluated according to the field of function. The concept of self-efficacy is divided into many different categories, such as social self-efficacy (SSE), academic self-efficacy (ASE), professional self-efficacy and emotional self-efficacy (ESE). When this concept is applied in an academic context, it is specifically referred to as ASE. ASE pertains to individuals' beliefs that they can successfully accomplish academic tasks at the desired level or attain certain academic goals (Pajares, 2007). In their meta-analysis of 59 studies, Honicke and Broadbent (2016) found a moderate relationship between ASE and academic performance, indicating that ASE is a highly relevant variable in educational studies.

Self-efficacy is also considered in the emotional dimension. ESE is related to the ability to evaluate one's own and others' emotional reactions (Choi et al., 2013). It can be defined as an efficacy belief that enables individuals to manage their negative emotions and strive to fulfill their goals under various circumstances (Bandura, 1993). Social self-efficacy refers to an individual's capacity to develop new friendships and establish social relationships (Gecas, 1989). In other words, SSE indicates one's perceived competence regarding social skills.

As can be seen, self-efficacy relates to specific situations, and individuals may exhibit high self-efficacy in some contexts and low self-efficacy in others simultaneously. However, it is clear that high levels of all three forms of self-efficacy contribute positively to an individual's self-perception and their perspective toward their environment. This suggests that an individual's self-efficacy can enhance their overall well-being.

Students with high self-efficacy tend to make more effort to overcome the difficulties they face on their own. Therefore, it is argued that self-efficacy is the starting point for the emergence of resilience in individuals (Everly et al., 2015). Resilience is an individual's ability to recover despite visible stressors (Egeland et al., 1993; Herrman et al., 2011). This concept has gained increasing importance in research in recent years due to the negative events experienced worldwide. It is also a multidimensional construct influenced by environmental context, cultural conditions and social factors (Connor and Davidson, 2003). In recent years, resilience has been considered a field-specific concept, with its different aspects – such as academic, emotional and behavioral resilience – being discussed (Jowkar et al., 2014). In the context of education, it can be said that AR is more prominent than other forms of resilience. AR is defined as "the heightened likelihood of success in school and other life accomplishments despite environmental adversities, brought about by early traits, conditions, and experiences" (Wang et al., 1994, p. 46). AR is an important variable that relates to educational outcomes and other psychological factors. For example, Mwangi et al. (2015) found a positive and significant relationship between AR and ACH in a study with 390 secondary school students. In a study involving 435 British undergraduate students, Cassidy (2015) found that ASE is related to AR and is a significant predictor of it.

Similarly, test anxiety is a variable frequently addressed in research. Bodas and Ollendick (2005) argue that test anxiety is a widespread problem across geographical and cultural boundaries.

Test anxiety is a specific type of anxiety. Anxiety reflects a future-oriented state of mind associated with being ready for possible impending negative events (Barlow, 2002). Test anxiety, on the other hand, refers to the phenomenological, physiological and behavioral reactions of individuals to possible negative results and failure in an exam or similar assessment situation (Sieber et al., 1977). Therefore, in the context where test anxiety is observed, there is a performance evaluation, which involves a more specific situation than general anxiety. Although test anxiety is a common problem for students, it is also emphasized as one of the most prominent sources of anxiety (Furr et al., 2001). Test anxiety consists of different components; in this study, we focus only on the cognitive component of test anxiety (Hembree, 1988), which is most consistently associated with academic performance. Cognitive test anxiety "consists of individuals' cognitive responses to evaluative situations or internal dialogues about evaluative situations before, during, and after evaluative tasks" (Cassady and Johnson, 2002, p. 272).

All three variables – test anxiety, self-efficacy and AR – play a role in affecting students' academic performance. Similarly, these variables are interrelated and influence each other. In their study with medical students, Hayat et al. (2021) examined the interconnections between self-efficacy, AR and test anxiety, showing that there were significant relationships among the three variables, with AR mediating the self-efficacy–test anxiety relationship. In another study, it was found that students with high self-efficacy reported lower levels of test anxiety and, therefore, achieved greater success (Elias, 2008). Therefore, students who are confident in their academic abilities are expected to have low test anxiety and high AR, and these variables can have a significant impact on students' ACH.

### Network analysis

Network analysis generates a network graph in which observed variables are represented by nodes (e.g., test and/or questionnaire items, psychopathological symptoms), and statistical relationships between nodes (e.g., partial correlations given all other nodes in the network) are depicted by edges (i.e., lines connecting the nodes). An edge between two nodes typically represents partial correlation coefficients, reflecting the remaining relationship between the two variables after controlling for all other variables (Epskamp et al., 2018). The edges can be either positive or negative, and the polarity of these relationships is shown graphically by using different colored lines: positive relationships are typically colored blue or green, while negative relationships are colored red (Hevey, 2018). Edges can also be weighted or unweighted. A weighted edge reflects the strength of the relationship between nodes by varying the thickness and color intensity of the edge connecting them: thicker, more intensely colored lines indicate stronger relationships. When the edge is unweighted, it can only represent the presence or absence of a relationship; in such a network, the absence of a relationship results in nodes not having a connecting edge (Hevey, 2018). The length of edges often indicates the strength of influence, with shorter edges suggesting a more immediate influence between nodes.

Using traditional analytical approaches in studies focusing on latent constructs can make it difficult to capture the details of interactions between variables (Isvoranu et al., 2016a). Therefore, alternative modeling strategies, such as network analysis, have been developed. Network models do not require a priori assumptions when defining dimensions but instead create a structure that emerges based on the data. This approach identifies the variables

that are central and most influential within this structure, facilitating effective interpretation of the findings with the help of powerful visualizations (Borsboom et al., 2021). These reasons have led to the frequent use of network analysis among researchers (Malas et al., 2024; McElroy et al., 2019; Wang et al., 2023; Zavlis et al., 2022).

### The present study

In this study, a network analysis was conducted on the variables of self-efficacy, AR, test anxiety and ACH, which are frequently preferred in educational research. This analysis focuses on the relationships between the variables and highlights their importance through statistical modeling (Gao et al., 2022). Network analysis is a data-driven approach; therefore, the relationships between the selected variables were explored without providing any prior information to the model. The analysis aims to make the findings more understandable through visual representation.

To our knowledge, there is no study that addresses all the variables we selected for our research simultaneously. Uygur et al. (2023) stated that ASE is an important determinant of desired academic outcomes in students. However, although academic outcomes and resilience are related to social–emotional competencies, studies focusing on SSE and ESE are more limited. As the importance of these three dimensions of self-efficacy is discussed separately in this study, it is expected to contribute to addressing this gap. In addition, network analysis is mostly used in medical and psychology studies and is less common in educational research conducted with student participants (e.g., Abacioglu et al., 2019; Dughi et al., 2023). Therefore, examining these variables, which are directly or indirectly related to students' well-being, together and with more innovative methods will contribute to the field. Our hope in conducting this study is to contribute to the integration of theoretical and applied research in the field of education and to provide researchers and practitioners with a better understanding of the interactions between students' cognitive and psychological characteristics.

### Methods

This study follows the steps recommended by Burger et al. (2023) for reporting network analysis studies.

### Participants and procedure

This study is a cross-sectional network analysis. The data used in this study were collected from seventh- and eighth-grade students, with the permission of Akdeniz University Ethics Committee. Important clarifications were given that the participants' identities would be kept private and that the information gathered would only be utilized for scientific research. Since the data were collected face to face under the teacher's verbal instructions and control, there were no careless response patterns or missing data in the dataset. The participants were first administered a questionnaire to collect demographic information, followed by the other scales. The time to complete the scales was approximately 15 minutes. A total of 828 Turkish secondary school adolescents participated in the study through convenience sampling method, 51.9% of whom were girls, and 44.1% were seventh-grade students. In terms of socioeconomic status, 8.5% of the students were in the low category, 84.8% in the medium category and 6.8% in the high category.

### Measures

#### Demographic form

A special questionnaire was designed and administered to students to obtain information on variables such as grade level, gender, socioeconomic status and city of residence.

#### The Self-Efficacy Questionnaire for Children (SEQ-C)

This scale is a five-point Likert-type scale (1 = not at all; 5 = very well) developed by Muris (2001) to measure the SSE, ASE and ESE of adolescents aged 14–17. When the psychometric properties of the scale were examined, it was determined that it consists of three sub-dimensions: ASE, SSE and ESE, with 7 items in each sub-dimension and 21 items in total (ASE: 3, 6, 12, 15, 17 and 20; SSE: 1, 5, 7, 10, 13, 16 and 18; ESE: 2, 4, 8, 11, 14, 19 and 21). As a result of the exploratory factor analysis, it was reported that these three factors explained 56.70% of the variance, and Cronbach's α internal consistency coefficient was .88 for ASE and ESE; .85 for SSE; and .88 for the total scale. A high score on the scale indicates that the adolescent has a high level of related self-efficacy. For this study, we included data collected on the Turkish version of the SEQ-C, adapted from Telef and Karaca (2012). In the present study, the three-dimensional structure was found to be acceptable, with the following fit indices: CFI = .91, TLI = .91, RMSEA = .058 (confidence interval [CI] 95% [.054, .063]), SRMR = .044, and reliable results were obtained (Cronbach's α for ASE = .84, SSE = .78, ESE = .80 and total scale = .90).

#### Cognitive Test Anxiety Scale—Revised (CTAR)

This scale is a 27-item scale developed by Cassady and Johnson (2002) to assess only the cognitive aspects of test anxiety. However, it was revised by Cassady and Finch (2014) to eliminate the reverse-coded items that were later determined to measure a separate construct and became the CTAR, a 25-item scale. The response categories to the scale items are 4 point (1 = *not at all like me*; 4 = *very much like me*). Cassady and Finch (2014) stated that the scale showed a unidimensional construct because of validity analysis. As the score obtained from the scale increases, it indicates that cognitive test anxiety increases. For this study, we included data collected on the Turkish version of the CTAR, which was adapted from Bozkurt et al. (2017). In the adapted scale, 2 items with insufficient factor loadings were discarded, and the scale was finalized with 23 items. In the present study, the unidimensional structure of the CTAR was found to be acceptable, with the following fit indices: CFI = .91, TLI = .90, RMSEA = .065 (CI 95% [.058, .072]), SRMR = .022, and reliable results were obtained (Cronbach's α for CTAR = .92).

#### The Academic Resilience Scale

This scale consists of six items developed to measure individuals' AR and includes a 7-point scale ranging from "not true of me at all" to "extremely true of me" (Martin and Marsh, 2006). The scale shows a unidimensional structure, and the Cronbach's α coefficient was reported as .89. For this study, we included data collected on the Turkish version of the ARS, which was adapted from Kapikiran (2012). In the present study, the one-factor structure was confirmed (CFI = .99, TLI = .98, RMSEA = .044 CI 95% [.023, .067], SRMR = .066), and reliable results were obtained (Cronbach's α for ASR = .82).

### Academic achievement

ACH refers to students' average achievement scores in all subjects at the end of the academic year. These scores were obtained from the school administration for each student. The ACH for the participants in this study ranged from 50 to 100 points.

### Data analysis

We carried out descriptive analysis in SPSS 24. Network analyses were performed with the bootnet (Epskamp and Fried, 2024), qgraph (Epskamp et al., 2023) and networktools (Jones, 2023) packages in the R program. First, the mean, standard deviation, skewness (Sk) and kurtosis (Ku) values of each item were analyzed. Since the Sk and Ku values were within the range of ±1, it was accepted that the responses to the items were normally distributed. Then, Pearson correlations between the total scores obtained from the scales and the arithmetic mean and standard deviation values were analyzed.

For the network analysis, we considered the three steps suggested by Epskamp et al. (2018, pp. 195–196): 1) estimation of the statistical model on data; 2) analysis of the network structure; and 3) assessment of the accuracy and stability of network parameters and measures. The pairwise Markov random field network model was used for the estimation and visualization of the networks. Under this model, a Gaussian graph model is preferred, where nodes represent the observed variables (here, items) and edges represent partial correlation coefficients between two variables after conditioning on all other variables in the dataset. Due to the ordered, nonnormal nature of the data, a polychoric correlation matrix was created by selecting "auto" for the correlation method. Since partial correlation coefficients may reflect spurious correlations that represent relationships that are not actually true (Epskamp and Fried, 2018), the graphical least absolute shrinkage and selection operator (gLASSO) regularization technique was used to address these spurious associations. This technique eliminates nonsignificant edges by estimating them as zero. The gLASSO regularization technique uses the extended Bayesian information criterion (EBICglasso) estimator, with the tuning parameter gamma set to 0.5, as in Isvoranu et al. (2016b). EBIC was chosen because EBIC produces better features in a graphical model environment than regular BIC when the tuning parameter is chosen appropriately (Foygel and Drton, 2010).

After all these selections, a graphical representation of the network was obtained, revealing the structural relationships between the nodes. For the analysis of this network structure, the weighted matrix and centrality indices were evaluated. The weighted matrix measures the relationships between nodes, and higher weights represent stronger relationships (Hevey, 2018). Centrality indices provide insight into the relative importance of the nodes in the network in determining the overall structure. Although strength, closeness and betweenness indices are used for the centrality index, in this study, we focus only on the results of the strength centrality index. Because it has been reported that obtaining stable results for betweenness and closeness can be problematic (Epskamp et al., 2018), they are often not reliably estimated (Mangion et al., 2022). The strength index expresses how strongly connected or conditionally related a given measurement node is, on average, to all other measurement nodes in the network. Centrality indices were calculated as standardized z-scores, with higher z-scores indicating higher centrality in the network.

*Network stability.* Subset bootstrapping was used to check for stability and accuracy (Epskamp et al., 2018). A 95%CI for edge-weight accuracy was calculated using a bootstrapped sample. The number of bootstrap samples was set to 1,000, and the bootstrap type was nonparametric. In nonparametric bootstrapping, observations in the data are resampled with replacement to create new plausible datasets (Epskamp et al., 2018, p. 199). A narrower CI indicates a more reliable network. Centrality stability was assessed through case-dropping subset bootstrapping (Epskamp et al., 2018). The ability to drop most cases from the dataset without significant changes in the centrality index of a node indicates that the network is stable (Gao et al., 2022, p. 4). In this method, various subsets of the sample are created, and the correlation between the centrality indices obtained from these subsets and the original centrality indices is examined. Epskamp et al. (2018) refer to this correlation as correlation stability (CS) and aim for a CS coefficient of no less than 0.25 and preferably above 0.5.

*Testing for significant differences.* Just because a node is more central does not mean it is substantially more central. Therefore, we use the bootstrapped difference test on the nonparametric bootstrap results to examine the stability of node strengths and edge weights.

### Results

### Descriptive statistics and correlations

The correlation matrix is presented in Table 1, with means and standard deviations shown in the bottom two rows. ASE was

**Table 1.** Correlation matrix and descriptive statistics

|           | 1       | 2       | 3        | 4        | 5       | 6      |
|-----------|---------|---------|----------|----------|---------|--------|
| 1. ASE    | 1.000   |         |          |          |         |        |
| 2. SSE    | .617*** | 1.000   |          |          |         |        |
| 3. ESE    | .662*** | .626*** | 1.000    |          |         |        |
| 4. CTAR   | −.340***| −.095** | −.216*** | 1.000    |         |        |
| 5. AR     | .610*** | .492*** | .484***  | −.313**  | 1.000   |        |
| 6. ACH    | .302*** | .137*** | .094**   | −.276*** | .311*** | 1.000  |
| M         | 23.828  | 24.264  | 22.336   | 48.298   | 19.916  | 84.070 |
| SD        | 5.844   | 5.894   | 5.868    | 13.601   | 5.537   | 11.882 |

*Notes*: ASE: academic self-efficacy from the sub-dimensions of SEQ-C, SSE: social self-efficacy from the sub-dimensions of SEQ-C, ESE: emotional self-efficacy from the sub-dimensions of SEQ-C, CTAR: Cognitive Test Anxiety Scale—Revised (23 items), AR: academic resilience scale (6 items), ACH: academic achievement.
**p < .01, ***p < .001.

significantly positively associated with SSE ($r = .617$, $p < 0.001$), ESE ($r = .662$, $p < 0.001$), AR ($r = .610$, $p < 0.001$) and ACH ($r = .302$, $p < 0.001$) and was significantly negatively associated with CTAR ($r = -.340$, $p < 0.001$) scores. Similarly, the other variables were significantly positively associated with each other, except for CTAR. The correlations among them ranged from small to medium.

### Psychometric network analysis (PNA)

The network resulting from the PNA, in which all items in the scales are included, is shown in Figure 1. The color of each node indicates to which scales/subscales the items belong. The width (i.e., thickness) of each line represents the strength of the relationship between different pairs of nodes, and the color represents the direction of the relationship (green: positive, red: negative). Of the 1,225 possible edge weights in the network of 50 nodes, 404 of them are nonzero (33%) weights, with a sparsity value of .670. Sparsity is a value between 0 and 1, and the higher the sparsity, the more weakly connected the network is (Molero et al., 2023). In this case, this network is somewhat weak. This result is expected in this network that includes different scale items as variables. In the predicted network, it was found that the items were clustered in accordance with their latent variables, as expected. When Figure 1 is examined, it can be seen that the items with high correlations are in the same scale/subscale. A clear distinction can be observed between the three clusters (CTAR, AR and SE). In particular, cognitive test anxiety is sharply separated from the other two positive psychological states: AR and self-efficacy. Moreover, the strongest relationships between items were observed within the scales themselves. In the ESE subscale, high correlations were observed between ESE8 (How well can you control your nerves?) and ESE11 (How well can you control your emotions?) ($r = .331$), and between ESE1 (How well can you express your opinions when your classmates disagree with you?) and ESE14 (How good are you at cheering yourself up when you are not feeling well?) ($r = .295$).

In the SSE subscale, high correlations were observed between SSE5 (How good are you at making friends with other children around you?) and SSE7 (How good are you at having a conversation with a stranger?) ($r = .234$), between SSE16 (How well can you explain a funny event to a group of students?) and SSE18 (How good are you at maintaining friendships with other children?) ($r = .212$) and between SSE5 and SSE18 ($r = .331$).

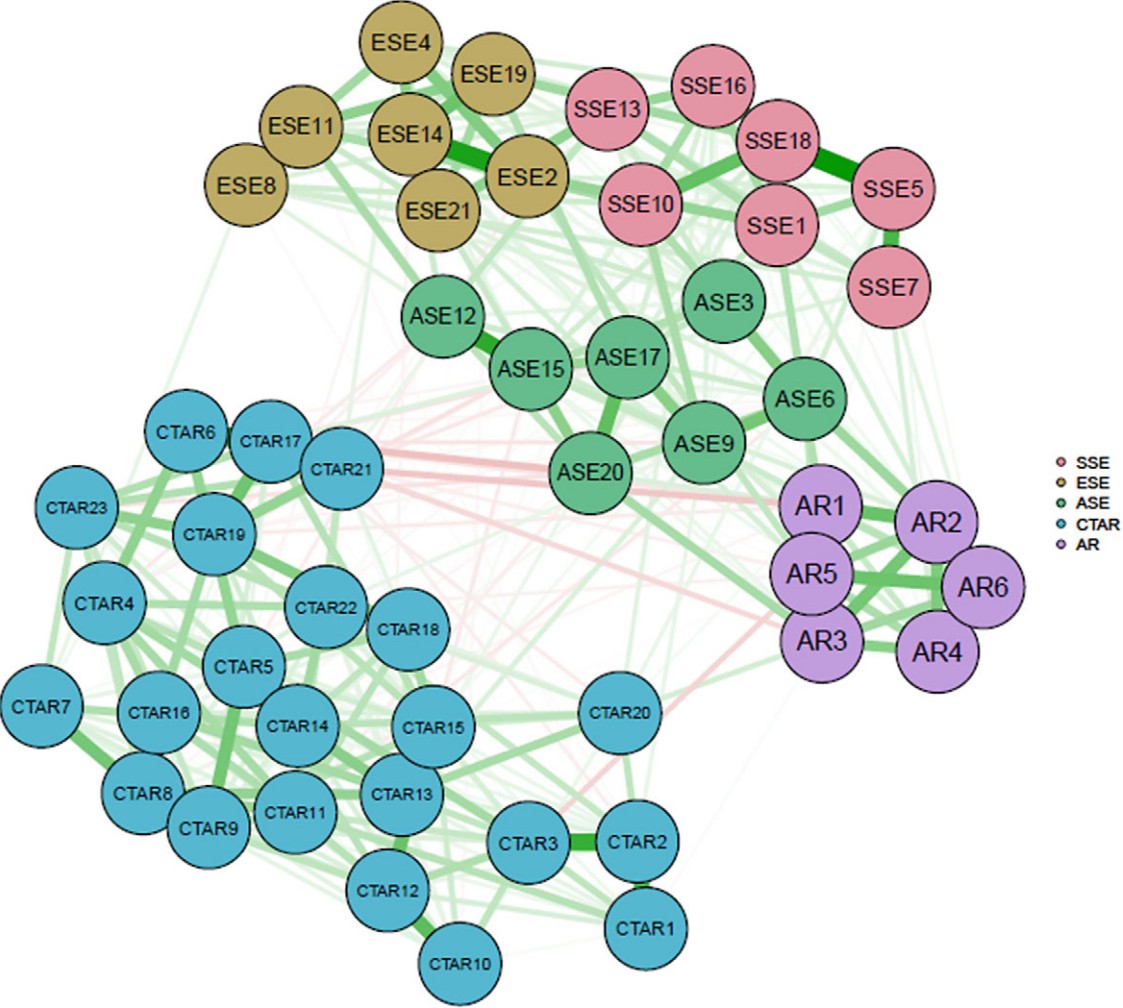

**Figure 1.** Psychometric network model of the scales. The nodes are labelled and colored according to the variable they theoretically represent. CTAR: Cognitive Test Anxiety Scale—Revised (23 items), AR: academic resilience (6 items), ASE: academic self-efficacy from the sub-dimensions of the Self-Efficacy Questionnaire for Children (SEQ-C), SSE: social self-efficacy from the sub-dimensions of SEQ-C, ESE: emotional self-efficacy from the sub-dimensions of SEQ-C.

In the ASE subscale, a high correlation was observed between ASE12 (How well are you able to concentrate in each of your classes?) and ASE15 (How well are you able to understand all of the lessons in school?) ($r = .272$), as well as between ASE17 (How well are you able to please your family with your studies in school?) and ASE20 (How well are you able to pass any exam?) ($r = .210$).

The high correlations between items in the cognitive test anxiety (CTAR) scale are as follows: CTAR6 (I am not good at exams) and CTAR17 (I cannot perform well on exams) ($r = .281$), CTAR17 and CTAR21 (after exams, I feel that I could have done better than I actually did) ($r = .271$), CTAR2 (I worry too much about doing well on exams) and CTAR3 (I am distracted by thoughts of failing while studying for exams) ($r = .261$) and CTAR1 (I lose sleep worrying about exams) and CTAR2 ($r = .244$).

On the AR scale, high correlations were noted between AR1 (I believe in my mental strength in exams) and AR2 (I work without giving up even in tasks that are difficult for me to accomplish) ($r = .188$), between AR5 (I do not let a bad grade affect my self-confidence) and AR6 (I am good at coping with failure at school, such as negative feedback on my homework or a bad grade) ($r = .198$) and between AR2 and AR3 (I am good at recovering poor grades in my courses) ($r = .168$).

Table A1 and Figure A1 in the Appendix show the strength values. Strength values are standardized *z*-scores, and any score greater than 1 (meaning that the average centrality of nodes is greater than 1 standard deviation) is considered high. High strength centralities were found for CTAR17 ($S = 2.321$), CTAR8 ($S = 1.858$), CTAR7 ($S = 1.804$), CTAR21 ($S = 1.365$), CTAR12 ($S = 1.302$) and CTAR15 ($S = 1.045$) in the CTAR scale; ASE15 ($S = 1.324$) and ASE6 ($S = 1.230$) in the ASE subscale; and SSE18 ($S = 1.116$) in the SSE subscale. In addition, the accuracy of edge weights was relatively reliable based on the results of the bootstrapped network analysis (Appendix Figure A2). We also analyzed the stability of the network and found an excellent level of stability for edge weight (.75) and for strength (.67) (Appendix Figures A3 and A4). The bootstrap difference test showed that most of the comparisons between edge weights and node strength were statistically significant (Appendix Figures A5 and A6).

### Network analysis

The network obtained from the network analysis, including the total scores of CTAR, AR scales and the subscales of SEQ-C (namely ASE, ESE and SSE), as well as ACH, is shown in Figure 2. A clear interrelationship was observed between almost all nodes in the network with six nodes. Of the 15 possible edge weights in the network, 14 are nonzero weights (93.3%) with a sparsity value of .067.

The network weight matrix and centrality measures for the total sample can be seen in Table 2. Especially strong connections emerge among ASE and ESE ($r = .381$), SSE and ESE ($r = .347$), ASE and AR ($r = .292$) and ASE and SSE ($r = .281$). Considering the strength values in Figure 2 and Table 2, ASE has the highest strength (rank) in the network. Therefore, this indicates that ASE has highly connected with other variables in the network, and the scores obtained from the ASE assessments can significantly affect other nodes.

### Accuracy and stability of network

To assess the accuracy and stability of our network, we first examined the edge-weight accuracy, specifically focusing on the

**Table 2.** The edge weight matrix in the network model and centrality measure (strength)

|       | ASE   | SSE   | ESE   | CTAR  | AR    | Strength |
|-------|-------|-------|-------|-------|-------|----------|
| ASE   | .000  |       |       |       |       | 1.643    |
| SSE   | .281  | .000  |       |       |       | .195     |
| ESE   | .381  | .347  | .000  |       |       | .410     |
| CTAR  | −.186 | .159  | −.049 | .000  |       | −.871    |
| AR    | .292  | .167  | .091  | −.134 | .000  | −.249    |
| ACH   | .166  | .000  | −.139 | −.165 | .161  | −1.128   |

*Notes:* ASE: academic self-efficacy from the sub-dimensions of SEQ-C, SSE: social self-efficacy from the sub-dimensions of SEQ-C, ESE: emotional self-efficacy from the sub-dimensions of SEQ-C, CTAR: Cognitive Test Anxiety Scale—Revised, AR: academic resilience scale, ACH: academic achievement.

estimated edge weights with 95% bootstrapped CIs, and the findings are shown in Figure 3. However, Epskamp et al. (2018, p. 200) state that edge-weight bootstrapped CIs should not be interpreted as significance tests regarding zero, and they recommend using them solely to demonstrate the accuracy of edge-weight estimates and to facilitate comparisons between edges.

Relatively large bootstrapped CIs indicate greater variability in the estimation of edge weights. Hence, high CIs indicate a degree of bias in the estimation of edge weights between two specific sets of nodes (Epskamp et al., 2018). In this study, very similar bootstrapped CIs were obtained for the estimated edge weights. This suggests that the edge weights are likely not significantly different from each other. Additionally, the bootstrapped 95% CIs of edge weights were narrow, indicating that the results of the network model were reliable.

Second, we estimated the stability of the centrality indices by analyzing network models based on subsets of the data. There is a slight decrease in edge and strength values (see Figure 4). Although this indicates that the stability of the results appears to be good, the numerical value for the CS should be checked. Epskamp et al. (2018) recommend that CS coefficients should not fall below .25 and preferably be above .50 for meaningful inferences. The edge stability coefficient and the centrality stability coefficient were .75, which can be interpreted as very good. Therefore, we conclude that the completeness order for the nodes is interpretable .

### Testing for significant differences

The resulting plots are presented in Figure 5. Panel A shows the results of the bootstrapped difference test for the edge weights. The edge weights between ASE and ESE are significantly different from those of almost all other variables ($p < .05$). The graph in Panel B indicates that most node powers are significantly different from each other. The node with the largest power, ASE, has significantly greater node power than all other nodes. ESE and SSE also have significantly larger node power than some of the other nodes. Thus, ASE and ESE are more centralized in the network, meaning these metrics have more influence on the other network variables.

### Discussion

The purpose of this study is to examine the relationship between self-efficacy, AR, cognitive test anxiety and ACH of secondary school adolescents within the framework of correlation network analysis. Although there are studies that examine these variables in

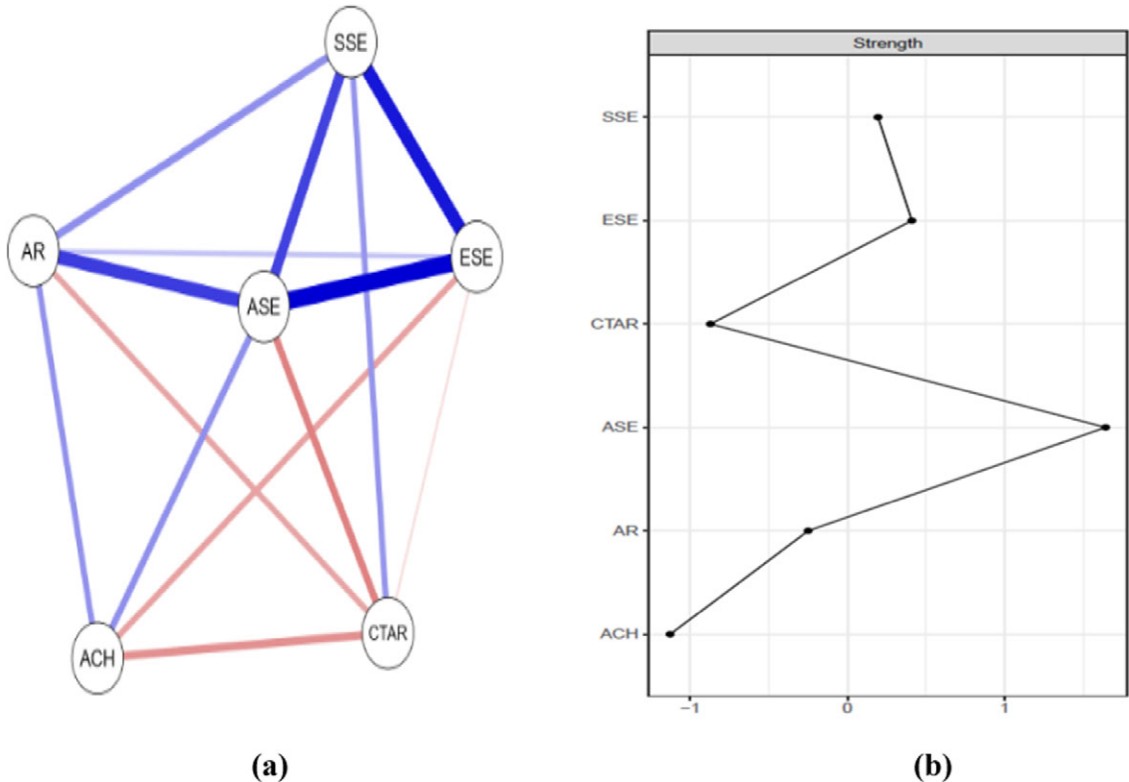

**Figure 2.** (a) Network plot. Positive edges are represented by blue lines and negative edges are represented by red lines. (b) Centrality plot of node strength.

different combinations, to our knowledge, this is the first study to examine the relationships among these variables together using network analysis.

In this study, the correlation values between the variables were first analyzed. As expected, the correlations between ASE, ESE and SSE in the same scale are high. Furthermore, it is clear that the variable with the highest and most significant correlation with other scale scores and ACH is ASE. Cognitive test anxiety shows significant and negative correlations with all other variables although its correlation with SSE is low.

Then, through the PNA, including the scale items, we examined how the items would be grouped without defining the latent variable, resulting in expected outcomes. Each item was clustered according to the latent trait to which it belonged, and while the items in the self-efficacy scale and those in the AR scale showed positive relationships, items from these scales generally exhibited negative relationships with items from the test anxiety scale.

Subsequently, a network analysis focused on the scale total scores was conducted, as these scores are central to the study. The network diagram obtained from this analysis shows that the strongest relationship is between ASE and ESE, followed by ASE and SSE. Won et al. (2024) also measured these three types of self-efficacy in students across two different time periods, finding that these three types of self-efficacy beliefs were positively correlated with each other, with the relationship between ASE and ESE being stronger than the relationships between ASE and SSE and ESE and SSE.

When the results related to the AR variable were analyzed, it was determined that the highest relationship emerged between ASE and AR, following the relationships among the self-efficacy subscales. However, according to the difference tests, there was no significant difference in edge weights between the ASE-ESE and ASE-AR

variable pairs. Previous studies have confirmed that these two variables are related to the academic status of students (Cassidy, 2015; Victor-Aigboidion et al., 2020; Wu et al., 2024). In addition, other studies have shown ASE to be the most important predictor of AR among ASE, ESE and SSE (see Uygur et al., 2023; Yıldırım and Kılıçaslan-Çelikkol, 2024). Although the relationship between the AR variable and the SSE variable was found to be stronger than that with ESE, there was no significant difference between the edge weights.

When the findings related to cognitive test anxiety are examined, the highest negative relationship with CTAR is observed with the ASE variable, followed by the AR variable. Consistent with previous studies, there is a significant negative relationship between test anxiety and ASE (Nie et al., 2011; Soltaninejad and Ghaemi, 2018) and between AR and cognitive test anxiety (Lei et al., 2021; Lim and Chue, 2023). A notable finding in the study is that the edge weight between test anxiety and SSE was found to be positive. This may be because test anxiety includes concerns about how it will be perceived by others.

When the findings of the study are analyzed in terms of the achievement variable, the relationship between achievement and ASE, and between achievement and AR, comes to the fore. When the literature is examined in the context of this finding, academic achievement/performance correlates with academic resilience (Choo & Prihadi, 2019; Rao & Krishnamurthy, 2018), with ASE (Afari et al., 2012; Coutinho & Neuman, 2008; Kitsantas & Chow, 2007; Eysenck et al., 2007; Zajacova et al., 2005), and with both academic resilience and ASE (Sadoughi, 2018) in which these variables are discussed together. Based on the results of this research, it can be said that increasing students' ASE and AR levels plays a notable role in enhancing their ACH levels. There is a negative relationship between achievement and test anxiety, which

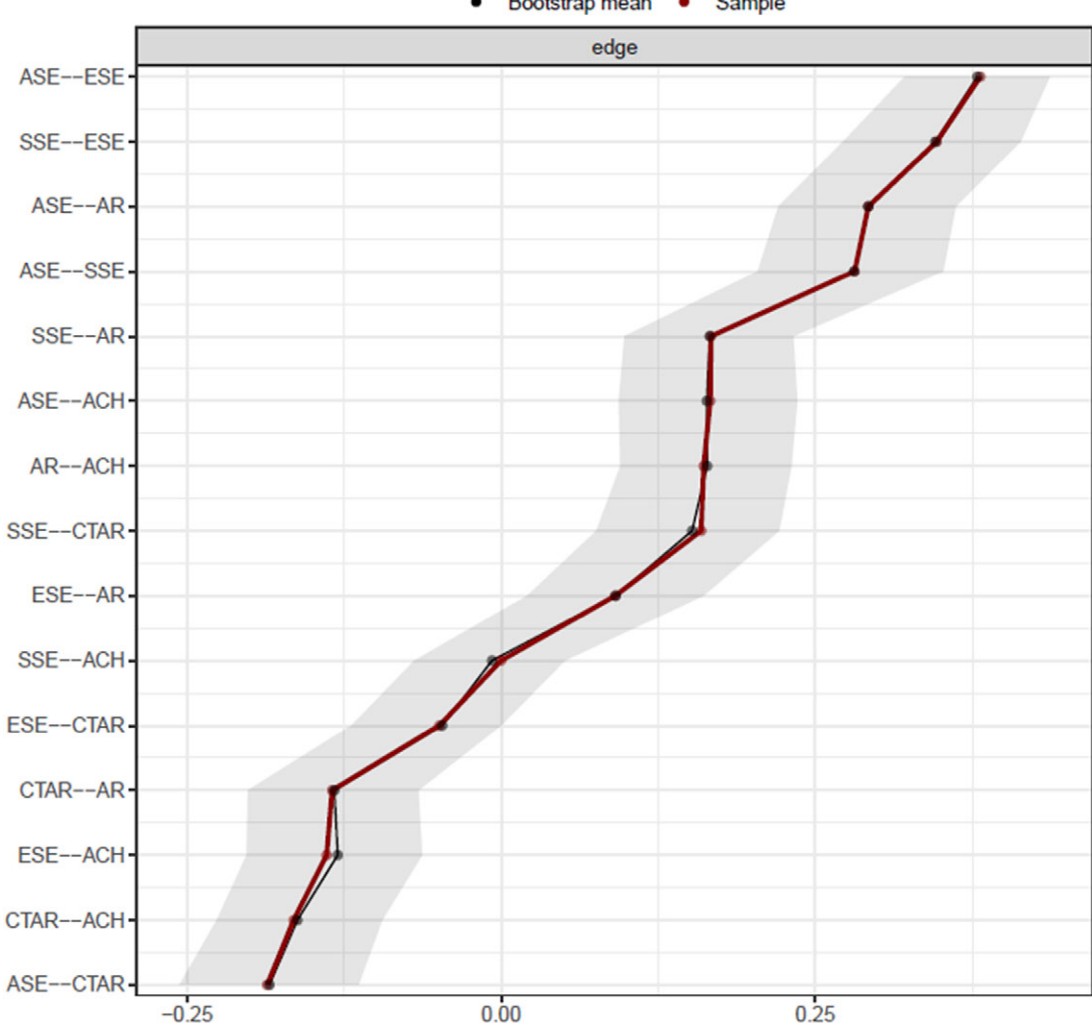

**Figure 3.** Network edge weight stability. Edge weights are shown with a red line, while 95% CIs around these edge weights are shown with a grey area. Bootstrapped mean 95% CIs are shown with a black line. ASE: academic self-efficacy from the sub-dimensions of SEQ-C, SSE: social self-efficacy from the sub-dimensions of SEQ-C, ESE: emotional self-efficacy from the sub-dimensions of SEQ-C, CTAR: Cognitive Test Anxiety Scale—Revised, AR: academic resilience scale, ACH: academic achievement.

aligns with the literature (Cassady, 2004; Cassady and Johnson, 2002; Thomas et al., 2017). While there is no relationship between achievement and SSI, there is an inverse relationship with ESI. This may be due to the cognitive factors involved in achievement.

The accuracy of the centrality indices in our network is quite good (CS = .75 for node strength). Similarly, the edge stability coefficient of .75 indicates that the depicted connections have a very good level of accuracy. The variable with the highest degree and strength of connectivity among the nodes in the network is ASE. ASE has a positive and significant relationship with all other analysis variables except CTAR, while the relationship with the CTAR variable is negative and significant. Therefore, the ASE variable has the most effective interactions with the other variables and is the most connected variable in the network graph. This variable is followed by ESE, SSE, AR, CTAR and ACH. However, ESE is not significantly more connected than SSE or AR, and CTAR and ACH variables do not differ from each other. Consequently, CTAR and ACH are the least connected variables in the network.

Through a network analysis approach, this study identified ASE as a central node that connects all the other variables in question. It also created a deeper understanding of how the studied variables are related to each other, both visually and through statistical results.

ASE has been emphasized as a notable variable in other studies and has been associated with students' perseverance and effort in the face of challenges (Bong and Skaalvik, 2003; Pajares, 2001; Won et al., 2024). Indeed, ASE has proven to be one of the strongest determinants of students' ACH (Cătălina et al., 2012; Richardson et al., 2012).

### Limitations, future directions and practical implications

The current study has some limitations. First, it used cross-sectional data and was conducted using a correlational design, which prevents the inference of causal relationships. Longitudinal or experimental studies may be more effective for understanding the relationships between these constructs in greater depth. Second, since the study was conducted only on middle school students, it does not provide insights into the relationships of these variables within the general population. Additionally, the data are based on self-report measures and may therefore contain some bias, especially in younger age groups. A similar study could be conducted with older university students. Furthermore, research on strategies to increase students' AR and ASE would also contribute to the literature. Another limitation is the problems arising from the difference

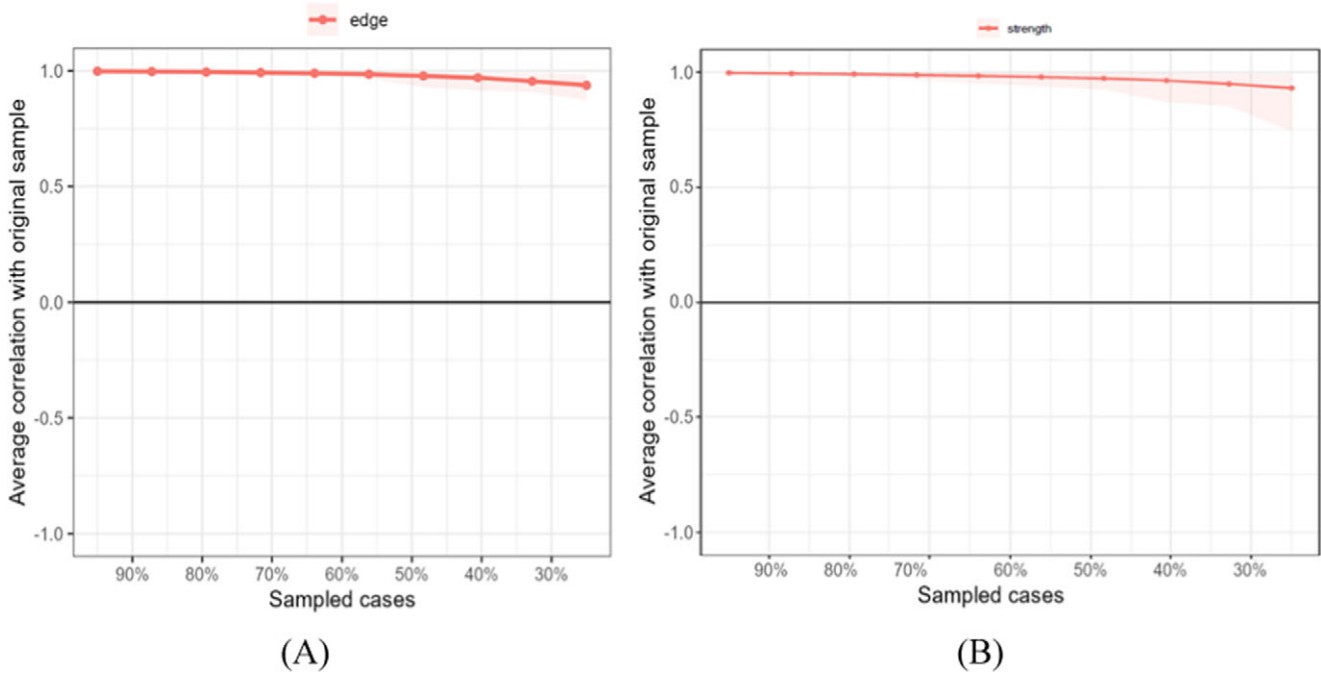

**Figure 4.** Case-dropping subset bootstrap. Panels A and B present average correlations in edge weight (A) and node strength (B).

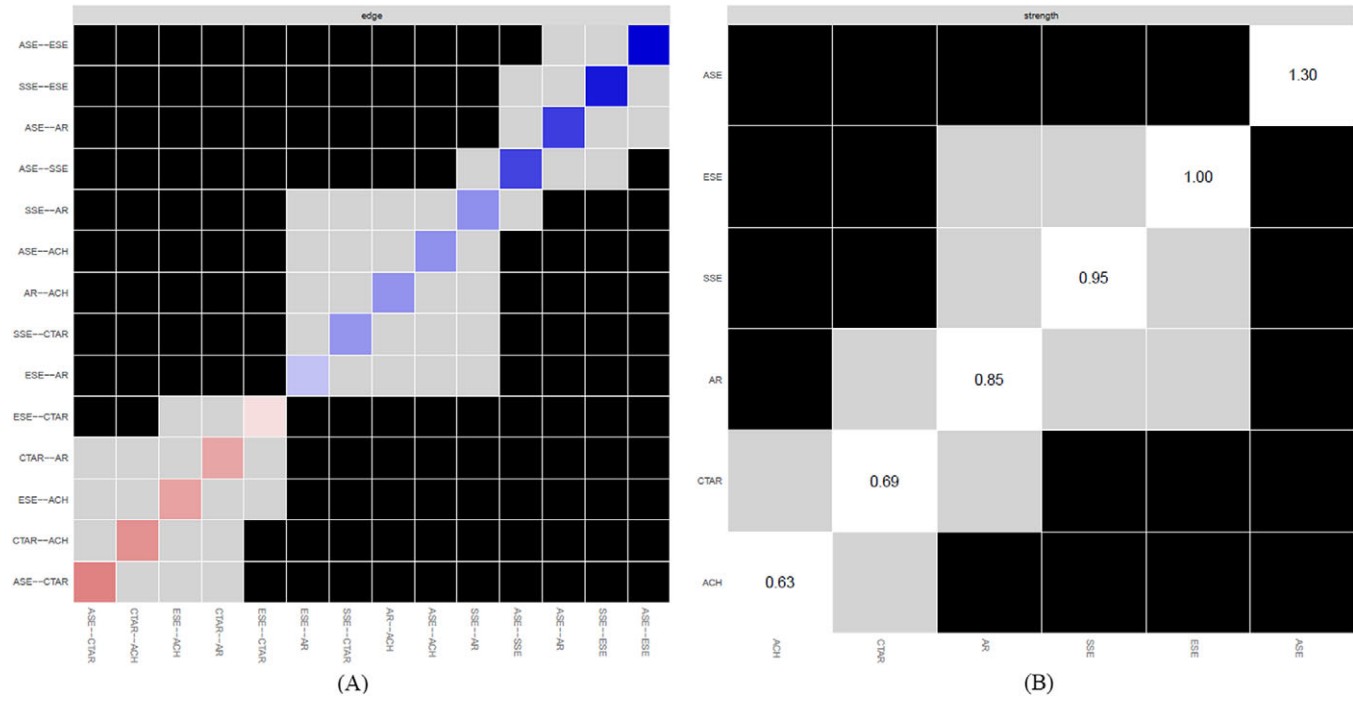

**Figure 5.** Bootstrapped difference test for edge weights (Panel A) and strength centrality (Panel B; α = 0.05). The color of the boxes indicates whether there is a significant difference (i.e., gray boxes reflect no significant differences and black boxes reflect significant differences). The colored boxes representing the diagonals in Panel A indicate the strength of the edge weight in the network graph (darker blue represents stronger positive connectivity, darker red stronger negative connectivity).

between the grades. The inclusion of school grades as ACH in this study includes differences in grading standards between schools. Other variables (e.g., emotional regulation) that may affect the variables selected for this study were ignored. New studies can be designed by taking these effects into account. This study provides several important recommendations for public policy makers and

academic institutes that help develop adolescents' ASE, which ultimately affects students' ACH. Teachers and other educational stakeholders should implement practices that support students' ASE. Seminars and sessions should be conducted to boost student AR and self-efficacy. The positive correlation between CTAR and SSE is a remarkable finding. It can be investigated whether similar findings

will be obtained in similar samples. In addition, this relationship can be examined with empirical studies. Future research could explore whether interventions targeting ASE lead to measurable reductions in CTAR over time, potentially clarifying causal pathways. Since it is thought that AR may be a potential mediator variable in the role between ASE and ACH, new research can be conducted on this issue.

## Conclusion

The finding that ASE is a highly connected variable in a network of multiple variables is important for improving students' academic lives. ASE helps students combat the obstacles they face, which can help increase their ability to cope with future challenges. Based on the results of this research, it can be concluded that students' ASE plays a crucial role in their psychological and academic well-being. At this point, researchers believe that it is important to include studies aimed at enhancing and maintaining students' ASE levels to improve their well-being. Additionally, researchers recommend that teachers adopt efforts to promote AR, as high resilience in students leads to many positive outcomes (Mwangi et al., 2015).

**Open peer review.** To view the open peer review materials for this article, please visit http://doi.org/10.1017/gmh.2025.17.

**Data availability statement.** The dataset used in the study can be sent to the readers by the corresponding author, if desired.

**Author contribution.** All authors contributed to the study conception and design. Manuscript writing and data analysis were performed by EYK; material preparation and data collection were performed by AS; manuscript writing and data preperation were performed by HaKo; manuscript writing and English editing were performed by HüKa; manuscript writing and editing were performed by GŞ. All authors read and approved the final manuscript.

**Financial support.** The authors have no relevant financial or nonfinancial interests to disclose.

**Competing interest.** The authors declare none.

**Ethics statement.** All procedures involving human subjects/patients were approved by the Social and Human Sciences Ethics Committee of the Akdeniz University with the number 803447.

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

## Appendix

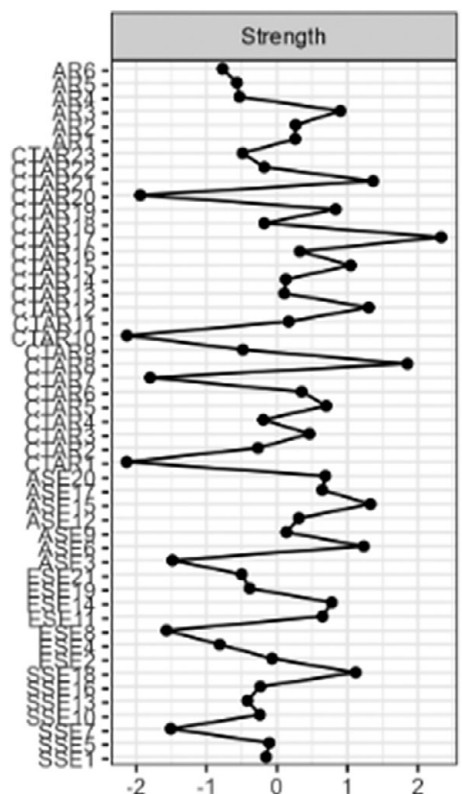

**Figure A1.** Node strength plot.

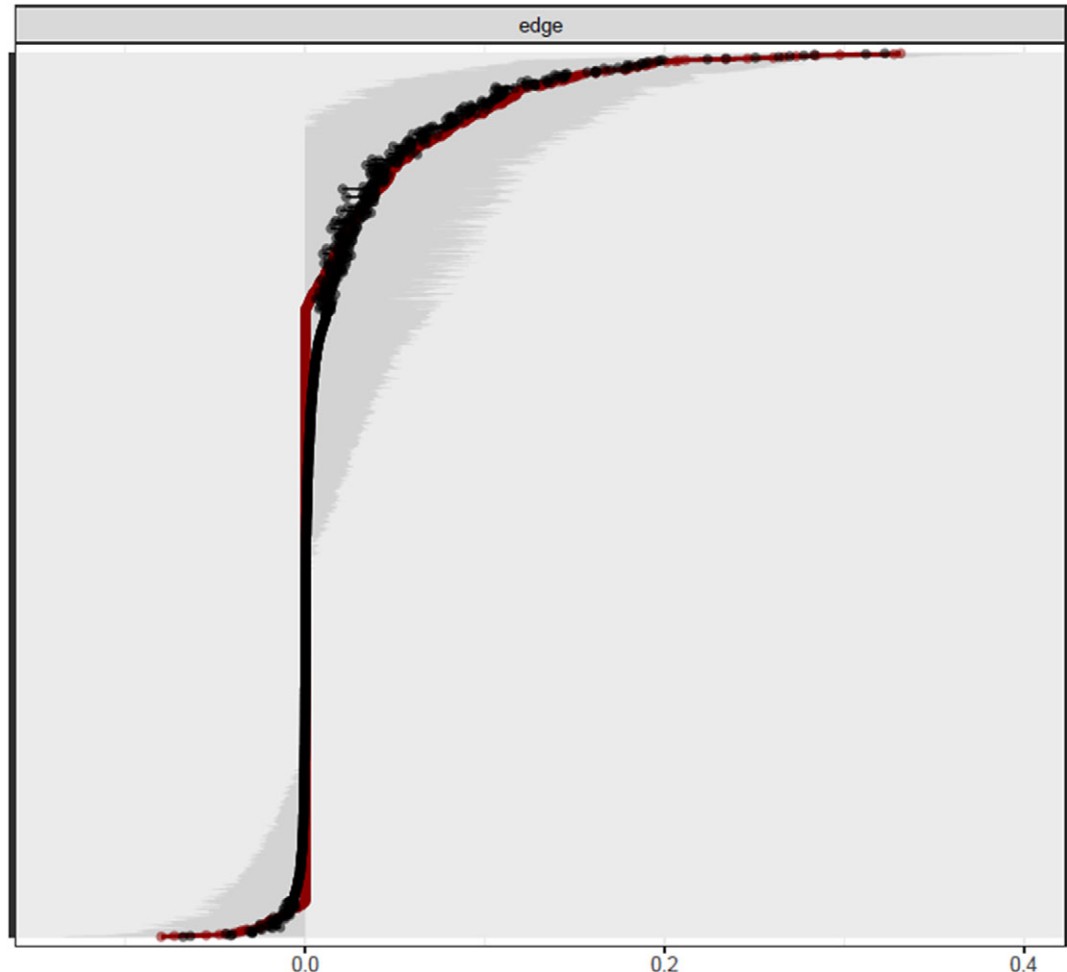

**Figure A2.** Accuracy of edge weights in item-level network.
*Note:* The horizonal area within the plot represents the 95% quantile range of the parameter values across 1000 bootstraps. The red dots depicts the sample edge weights, while the black dots indicate the bootstrap mean values. and the gray bar depicts the bootstrapped confidence interval.

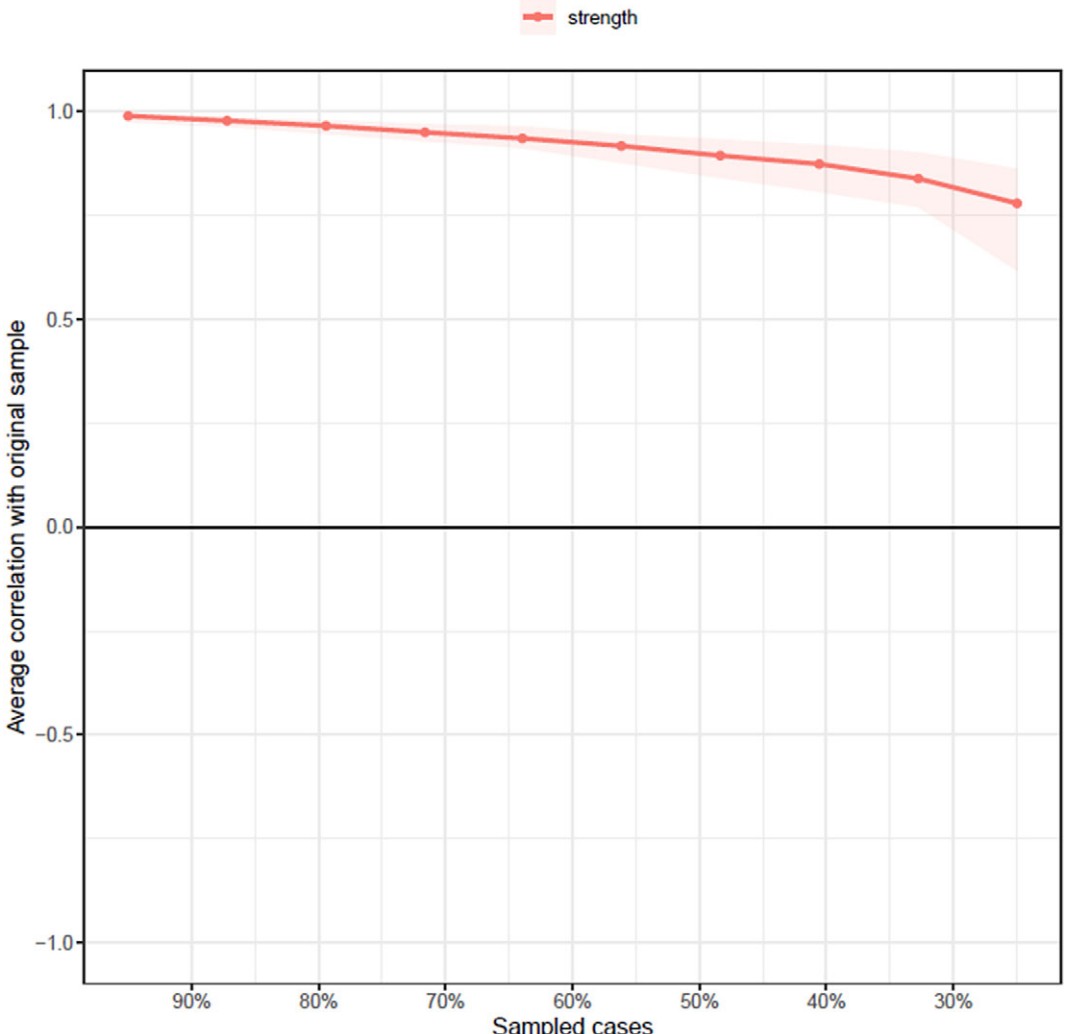

**Figure A3.** Accuracy and stability of strength in item-level network.
*Note:* Average correlation between node strength sampled with persons dropped and the original sample.

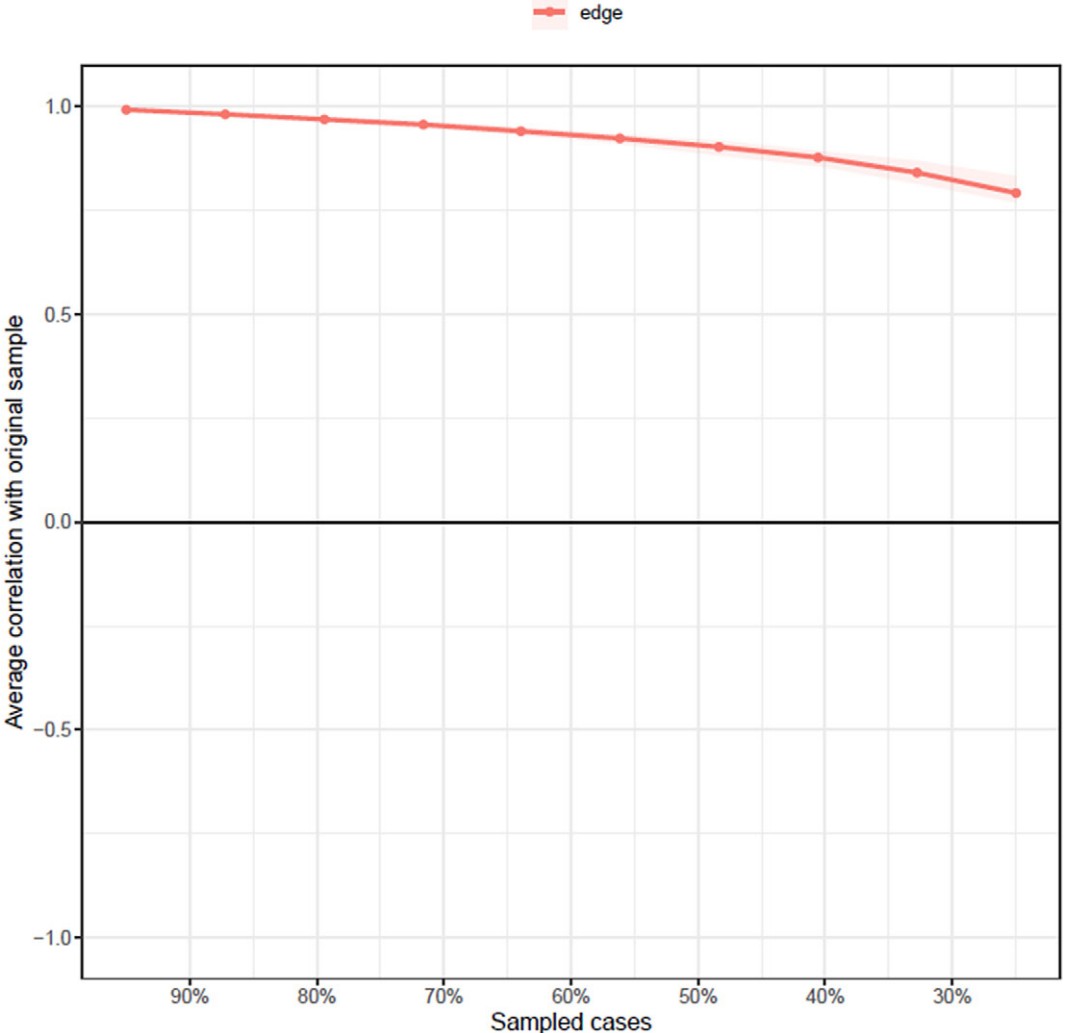

**Figure A4.** Accuracy and stability of edge weight in item-level network.
*Note:* Average correlation between edge weight sampled with persons dropped and the original sample.

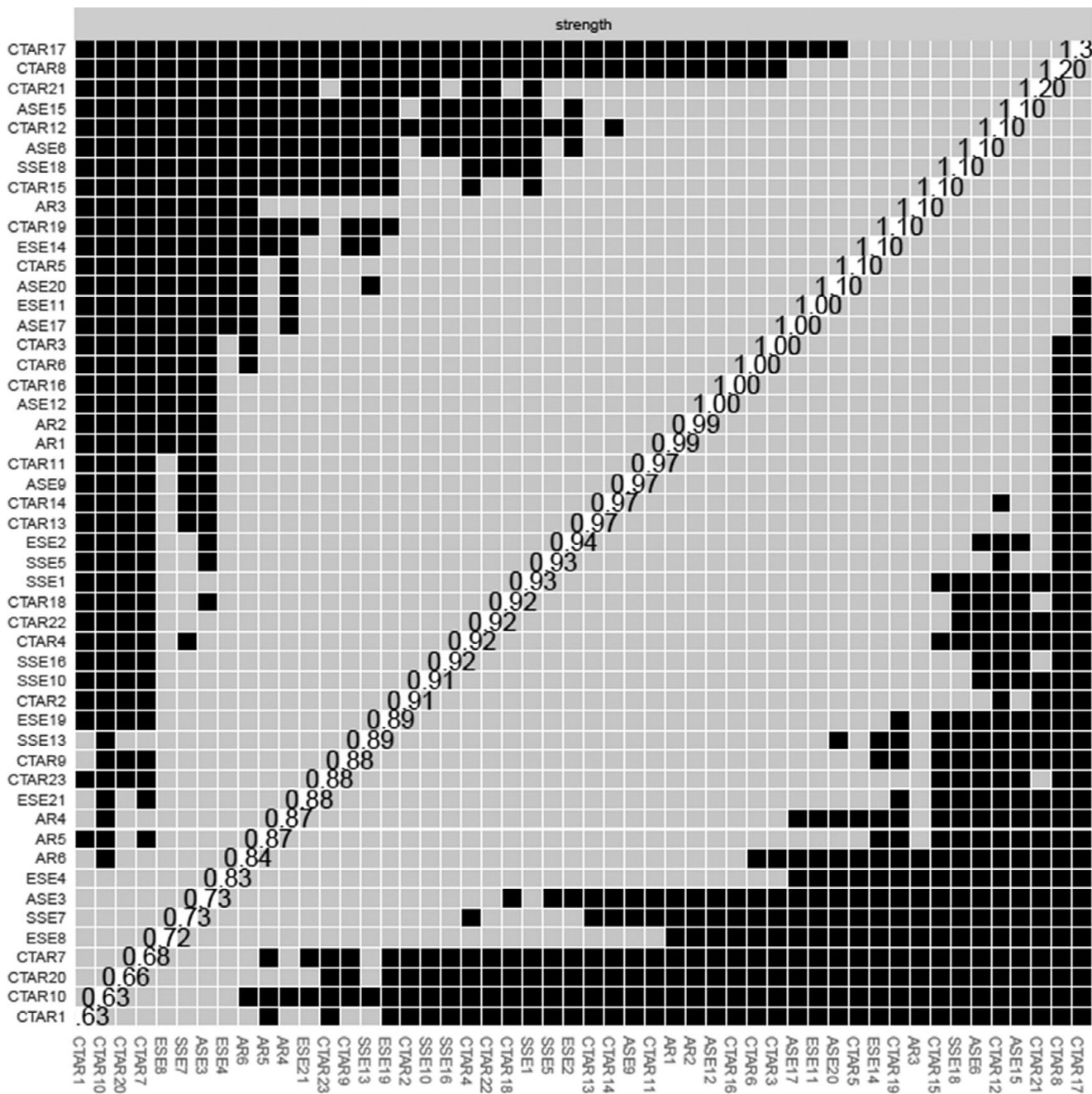

**Figure A5.** Bootstrapped difference test for node strength in item-level network.
*Note:* Gray boxes indicate node strength that do not differ significantly from one another, while black boxes indicate node strength that do differ significantly.

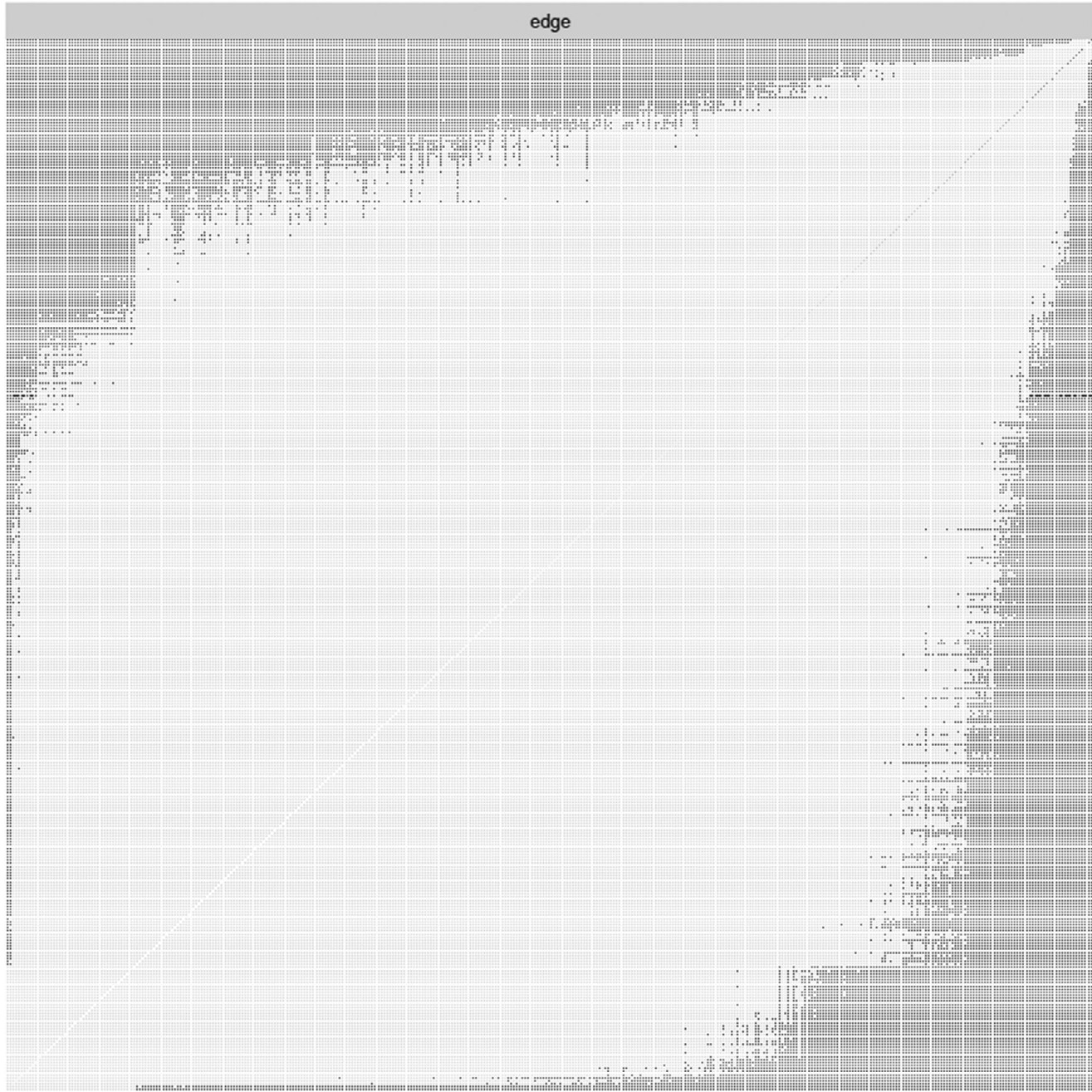

**Figure A6.** Bootstrapped difference test for edge weights in item-level network. The significance difference testing (α=0.05) examines whether edges significantly differ from each other in strength. The color of the boxes indicates whether there is a significant difference (i.e., grey boxes reflect no significant differences and black boxes reflect significant differences).

**Table A1.** Centrality measures per variable in the item-level network

| Variable | Strength |
| --- | --- |
| ASE3 | −1.48552312 |
| ASE6 | 1.22969597 |
| ASE9 | 0.13253279 |
| ASE12 | 0.30886358 |
| ASE15 | 1.32371862 |
| ASE17 | 0.63964156 |
| ASE20 | 0.68641511 |
| SSE1 | −0.15642327 |
| SSE5 | −0.10989737 |
| SSE7 | −1.50838502 |
| SSE10 | −0.24328013 |
| SSE13 | −0.42370687 |
| SSE16 | −0.23649484 |
| SSE18 | 1.11560358 |
| ESE2 | −0.07308741 |
| ESE4 | −0.81409745 |
| ESE8 | −1.57125949 |
| ESE11 | 0.64350047 |
| ESE14 | 0.77551959 |
| ESE19 | −0.38881683 |
| ESE21 | −0.50260123 |
| CTAR1 | −2.13661743 |
| CTAR2 | −0.27089017 |
| CTAR3 | 0.46043870 |
| CTAR4 | −0.19730320 |
| CTAR5 | 0.69826696 |
| CTAR6 | 0.34919318 |
| CTAR7 | −1.80385817 |
| CTAR8 | 1.85118372 |
| CTAR9 | −0.48403743 |
| CTAR10 | −2.13215618 |
| CTAR11 | 0.16370669 |
| CTAR12 | 1.30179318 |
| CTAR13 | 0.10451068 |
| CTAR14 | 0.12466414 |
| CTAR15 | 1.04597309 |
| CTAR16 | 0.32219512 |
| CTAR17 | 2.32970516 |
| CTAR18 | −0.18430632 |
| CTAR19 | 0.83123796 |
| CTAR20 | −1.94314492 |
| CTAR21 | 1.36478936 |
| CTAR22 | −0.18460605 |

(*Continued*)

**Table A1.**  (*Continued*)

| Variable | Strength |
| --- | --- |
| CTAR23 | −0.48868593 |
| AR1 | 0.25619289 |
| AR2 | 0.26126395 |
| AR3 | 0.90037627 |
| AR4 | −0.53616523 |
| AR5 | −0.56992960 |
| AR6 | −0.77570865 |

*Notes:* ASE: academic self-efficacy, ESE: emotional self-efficacy, SSE: social self-efficacy, CTAR: cognitive test anxiety, AR: academic resilience.