## [Reviewer Report]

This study explores the relationships among self-efficacy, academic resilience, cognitive test anxiety, and academic achievement in Turkish secondary school students using a network analysis approach. Data were collected from 828 students through validated self-report measures and academic records. The analysis, employing EBICglasso, revealed that academic self-efficacy (ASE) plays a central role, showing positive associations with academic resilience and achievement but a negative association with cognitive test anxiety. Academic resilience was identified as a mediator between ASE and achievement, while cognitive test anxiety negatively influenced these connections. These findings highlight ASE’s pivotal influence on students’ academic well-being and underscore the need for targeted interventions to enhance self-efficacy, reduce anxiety, and build resilience. The study provides valuable insights for educators and policymakers to design effective strategies for improving academic outcomes. Below, my feedback has been provided:

The title reflects the main themes of the research and the methodology used. It is informative and sets appropriate expectations for the content. However, you might consider making the title slightly more concise and engaging.

The abstract is effective but could be slightly refined for conciseness and emphasis. The sentence “The results highlighted the crucial and pivotal role of academic self-efficacy” could be simplified to avoid redundancy.

Including a brief mention of the study’s practical implications in the final sentence might further strengthen the abstract’s impact.

The introduction is well-written but could benefit from slight refinements to enhance clarity and depth. Expanding slightly on why network analysis is particularly suitable for this research question might provide additional context.

The methodology is comprehensive, detailing the sample, data collection process, and analytical approach. The use of EBICglasso is innovative. It might be helpful to provide a brief explanation of why EBICglasso was chosen over other methods, especially for readers unfamiliar with this approach.

Consider potential limitations of the method, including its dependence on assumptions of Gaussian distribution and its sensitivity to sample size. Including a note on whether the scales’ psychometric properties were verified within the study sample or adopted from prior research might further clarify the methodology.

The discussion of centrality measures, particularly the dominance of academic self-efficacy, could be expanded to include practical implications. If feasible, providing additional visuals or zoomed-in views of specific clusters in the network might enhance clarity for readers. The results are clear and compelling, with room for slight elaboration on key findings.

It might strengthen the discussion to further compare your findings with prior research, particularly on academic resilience and cognitive test anxiety. Consider elaborating on potential interventions that could leverage self-efficacy to mitigate test anxiety, as this could provide practical value to educators.

Avoid overusing phrases like “crucial” or “pivotal” to maintain the strength of your arguments. Avoid overusing phrases like “as seen” or “it is evident,” which can feel redundant.

A brief mention of how participant anonymity was maintained (e.g., coding of data, secure storage) would enhance the transparency of the ethical process.

Some parts feel repetitive. For example, multiple citations are provided for well-established relationships like self-efficacy and achievement. You might consolidate these into a single statement.

Consider focusing more on gaps in the literature, as this is where the paper makes its most valuable contribution.

The theoretical explanation could be shortened without losing depth. Introduction section could benefit from restructuring to reduce length and improve focus.

It may be beneficial to elaborate on how participants were recruited. For instance:

-Were schools randomly selected?

-Were students chosen through stratified sampling to ensure representation of different socioeconomic statuses?

The statement “The data set did not include careless response patterns or missing data” is positive, but the procedures used to identify and exclude careless responses could be explained in more detail.

Although the fit indices (e.g., CFI, TLI, RMSEA) are provided, a brief comparison to benchmark values (e.g., “acceptable thresholds”) could enhance interpretability for readers.

It may be helpful to justify why all three subdimensions (academic, social, emotional) were included in the analysis, especially if the focus of the study is primarily on academic outcomes.

Consider discussing why the cognitive component of test anxiety was prioritized over other components (e.g., emotional or behavioral). This choice aligns well with the study’s goals, but explicitly stating the rationale would strengthen the methodology.

While the psychometric properties are strong, the theoretical justification for including resilience as a variable in the network model could be expanded.

Discuss any potential biases that might arise from relying solely on school-reported grades (e.g., variations in grading standards).

The description of bootstrapping could be expanded to clarify how it ensures the reliability of edge weights and centrality measures.

A brief explanation of why expected influence (an alternative centrality measure) was not considered might preempt potential critiques. This would enhance the methodological clarity of the study.

While correlations are reported, the rationale behind their importance in the context of the study could be briefly elaborated. Consider including effect size interpretations (e.g., small, medium, large correlations) to help readers gauge the practical significance of the relationships.

The authors could elaborate on the implications of the sparsity value (e.g., does it indicate a highly connected or relatively sparse network compared to other studies?).

More interpretation of the edge weights would enhance the results' narrative.

Discuss any unexpected relationships. For instance, the positive relationship between CTAR and SSE might warrant further explanation or hypotheses.

Consider discussing the limitations of bootstrapping in the context of the dataset, such as potential biases introduced by sampling procedures.

The positive edge between CTAR and SSE is an intriguing finding that warrants further discussion. Hypotheses about social pressures or the role of interpersonal dynamics in test anxiety could be explored.

Consider mentioning any interventions or strategies that might address CTAR’s negative impact on other variables.

The implications of AR as a mediator between ASE and achievement could be discussed in greater depth.

Adding annotations or brief explanations directly on the figures could make them more user-friendly for readers unfamiliar with network analysis.

The positive association between CTAR and social self-efficacy (SSE) is briefly mentioned but requires further exploration. This relationship could be hypothesized as reflecting the impact of peer interactions or pressure.

Consider elaborating on AR’s potential mediating role between ASE and achievement.

The comparison with previous studies could be more detailed. For instance, directly addressing how the study’s findings confirm, extend, or contradict earlier work would strengthen its contribution to the field.

The practical applications of ASE’s centrality could be elaborated further.

Suggesting specific policies or classroom practices that align with these findings would make the implications more actionable.

The implications for addressing CTAR are somewhat underdeveloped.

The discussion of limitations could be expanded to include:

-Potential biases in self-reported measures, such as social desirability effects.

-The generalizability of findings to other cultural or educational contexts, given the Turkish sample.

-The limited scope of variables, as other factors (e.g., emotional regulation, peer influences) may also play significant roles.

Expanding on specific research questions for future studies would add depth. For example:

“Future research could explore whether interventions targeting ASE lead to measurable reductions in CTAR over time, potentially clarifying causal pathways.”

Exploring cultural differences in the network structure of these variables could also be an intriguing direction.

Suggesting methodological advancements, such as the inclusion of dynamic network analysis, could strengthen the forward-looking aspect of the discussion.

Your study provides significant findings within the context of Turkish students. However, to enhance the global relevance of your work, it may be beneficial to reference studies conducted in different geographical and cultural contexts or compare your findings with those from such contexts. For example, including examples from non-Western or cross-cultural studies could strengthen the international significance of your results.

---

## [Reviewer Report]

I am intrigued by your dataset, and I find the overall study to be of supreme interest. My own work intersects heavily with your data collection efforts, and I’m excited that you have data from a valuable age (one that has been significantly under-explored). I believe that the dataset has great promise, and that the study will reach publication - but in the current form I have reservations that I will share. The reservations are really more a matter of opinion and suggestion - nothing specifically untoward or inappropriate to report - but if I were the author or the editor, I would enact significant revisions to the analyses, literature review, and discussion.

The primary are of concern that I will express is a difference of opinion in the value of network analyses. I will disclose my bias on this matter openly, to support the ability of the editors and authors to come to a decision. Essentially, I find network analyses to be a clever tool to boost the analytic potential for largely unknown networks of data (and particularly for large datasets). They provide the opportunity to explore data and identify possible “clumps” in the response patterns that may prove more interesting and valued. They are also - again...just my opinion...seriously lacking in value for deep theoretical development or exploration. They are essentially (my opinion) weak alternatives to more advanced and probabilistic models that would be better suited in a study as valuable as this one - where we already know a good deal about the variables in hand. That is - I believe that the offering of a network analysis for these data is a mistreatment and waste of a great dataset - and I would advocate for a structured equations model approach that predicted achievement from a mediated/moderated model that employs these variables. The authors claim that no prior studies have worked with these variables (it’s perhaps true because of the broad measure of SE that has been used in this study...but I can think of 3 of my own studies that approximate these variables that would suggest your model should bear significant benefit to the field). In the end, I believe you will be able to demonstrate a model that illustrates that self-efficacy is a precursor variable that negatively predicts TA and positively predicts resilience. The path from TA to achievement will likely be negative (or small) and the path from resilience to achievement will be positive. The total effects will favor ASE as the most powerful predictor (with both indirect and direct effects showing value. A nice little visual model will likely arise. As a personal preference, I would follow up with analyses focused on non-linear relationships (my own work with SE and CTA in particular has demonstrated that TA is highest with there is greater “uncertainty” (not low SE...but unsure).

If the decision is to keep the network analysis as the preferred statistical offering, I would be very clear that there is very little additional information gained in this study beyond what is offered in the correlation table. That is - we know based on that table all the things that your scale-level network analysis offers. I am personally interested in the network analysis of the individual items for the 5 main variables (in particular the CTAR items) - but most people will likely find that information to be not terribly valued.

As for the literature review - I found the presentation to be somewhat dated in most areas, and I found that the presentation of the information on the variables was dominated by single-variable definitions for the most part. If it’s true that the variables have been studied together (at least in different combinations...which they have), that information should be evident here. We know that TA and SE are negatively correlated (and that there is a curvilinear relationship found in some studies...when they check for it). We know that resilience and TA are related (negatively - see Putwain for instance - he very clearly explained this relationship). We also know that there is a clear negative relationship between TA and most academic measures...but the achievement variable used in this study is perhaps not as reliable a measure as some other studies. That is, we see more durable relationships on standardized tests because they are less likely to be influenced by various biases in assigning grades (which appears to be the option here). More information on the achievement variable may clarify my thinking on this.

For the discussion, again - if using only network analyses, I believe pulling back from “the most important variable” when talking about SE is important. It is highly connected in this network model - that’s reasonable. However, to claim it is the most important variable affecting student well-being is a gross overstatement of the data (and not actually measured in this study in my opinion).

---

## [Reviewer Report]

Thank you for the opportunity to review your study. Your research, which examines the relationships between academic achievement and psychological characteristics of secondary school students using network analysis, makes a valuable contribution to the field. The findings are particularly insightful and have the potential to inform educational policies. Additionally, the use of visual representations strengthens the clarity of the findings, making the study more accessible to readers. I also believe that the revisions made throughout the manuscript have been positive and have further improved the overall quality of the paper. Congratulations on this important work, which provides a solid foundation for future research in education and psychology.

Best regards.